# Self-Repairing Composites for Corrosion Protection: A Review on Recent Strategies and Evaluation Methods

**DOI:** 10.3390/ma12172754

**Published:** 2019-08-27

**Authors:** Poornima Vijayan P, Mariam Al-Maadeed

**Affiliations:** 1Department of Chemistry, Sree Narayana College for Women, Kollam, Kerala 691001, India; 2Center for Advanced Materials, Qatar University, Doha 2713, Qatar; 3Materials Science and Technology Program, Qatar University, Doha 2713, Qatar

**Keywords:** protective coating, self-healing, anticorrosion

## Abstract

The use of self-healing coatings to protect metal substrates, such as aluminum alloys, stainless steel, carbon steel, and Mg alloys from corrosion is an important aspect for protecting metals and for the economy. During the past decade, extensive transformations on self-healing strategies were introduced in protective coatings, including the use of green components. Scientists used extracts of henna leaves, aloe vera, tobacco, etc. as corrosion inhibitors, and cellulose nanofibers, hallyosite nanotubes, etc. as healing agent containers. This review gives a concise description on the need for self-healing protective coatings for metal parts, the latest extrinsic self-healing strategies, and the techniques used to follow-up the self-healing process to control the corrosion of metal substrates. Common techniques, such as accelerated salt immersion test and electrochemical impedance spectroscopy (EIS), for evaluating the self-healing process in protective coatings are explained. We also show recent advancements procedures, such as scanning vibrating electrode technique (SVET) and scanning electrochemical microscopy (SECM), as successful techniques in evaluating the self-healing process in protective coatings.

## 1. Introduction

It is important to protect the surface of metals against corrosion. This protection is more important when compared to the bulk protection, as the surface is more exposed to the environment and prone to mechanical and solvent attacks. Corrosion can be expected in the metal if there are cracks or pits on the surface. The estimated cost of corrosion is approximately 3.4% of the global GDP (2013) [1]. Most of this cost is related to the inspection processes, repairing techniques, and environment and safety procedures. One of the most expensive corrosion cost is in oil and gas facilities. Protection against corrosion is important for the modern world, as it will reduce the financial cost. It is important to ensure the safety and reliability of industry, reduction of waste production, and energy consumption.

Cathodic protection, anodic protection, addition of inhibitors, protective coating, etc., are commonly practiced corrosion protection methods. Among them, a widely accepted protection method is the use of a suitable coating, which can protect the surface of the metal from the environment. Chemical substances, called inhibitors, can be added into coating in small quantities to decrease the corrosion rate. The basic idea of coating is to prevent the diffusion of oxygen and moisture to the metal. In addition to that, organic coatings have excellent impact resistance, aesthetic properties, and substrate adhesion [2]. Among protective coatings, organic coating is the oldest and most widely used protective coating technology [3]. Inhibitors can be added to the coatings, which ensures that they will not be leached and it is effective for the used metal [4].

As a variant of organic coating, organic-inorganic hybrid coatings consist of both the organic component and inorganic fillers (mostly silica), bonded together by the covalent bonds [5]. Hybrid coatings with titania as the inorganic component have also been reported in sol-gel coating films and nanocomposites coatings [6,7]. The combination of these two types of materials will ensure good mechanical properties, flexibility, adhesion, and transparency. Though organic coatings are effective, the needs for green anticorrosion techniques are high due to environmental issues such as greenhouse gas emission and global warming. Moreover, the economic concerns of crude oil and manufacturing prices encourage us to look to green alternatives for organic coating.

With the advancement in coatings research, many researchers have reported on smart coatings that show stimuli responsive action. The smart coating can include different responses, such as self-healing, self-cleaning, corrosion sensing, and anti-fouling. Among smart coating technologies, the self-healing coating concept has been introduced as a novel approach for achieving the corrosion protection function. The automatic response to corrosion can be tailored to different factors, such as abrasions, change in pH, surface tension, and temperature. Self-repairing organic and inorganic coating can increase the functional lifetime of the resulting coating structure. The self-healing functionality can be imparted, due to the presence of nano/micro containers of polymer and inorganic origin in the coating structure. The current review paper intended to detail these extrinsic strategies in designing self-healing coating structures with greater focus on green approaches. Depending on the end use applications, the follow-up methods that were adopted to evaluate the healing process are important and those will also be discussed.

## 2. Self-healing Strategies in Protective Coatings

### 2.1. General Methods 

With the introduction of self-healing concept, different self-healing strategies have been adopted to repair the cracks and damages in the coating systems [8]. Intrinsic and extrinsic strategies have both been applied to develop self-healing coatings. In intrinsic self-healing, the crack recovery can be achieved with reversible chemistry [9]. In the extrinsic type, either a capsule or vascular network aided healing method is used. In capsule aided self-healing, the healing agents are encapsulated in suitable capsules for delivering the healing agent into the damaged area [10]. Alternatively, a coating consisted of vascular network can be used to provide a uniform healing system throughout the surface [11].

Self-healing coatings that are based on micro/nano capsules are the most practiced systems. The factors, such as controlled uptake and release of healing agent/corrosion inhibitor, high loading capacity, multi-functionality, etc. can control the efficiency of such capsules. Microcapsules having a shell of polymer origin, such as epoxy, urea–formaldehyde (UF), and polyurethane (PU) are commonly used with either healing agents or corrosion inhibitors as the core materials [12,13,14]. Apart from microcapsules, nanocapsules, such as silica nanocapsules [15], ceramic nanocontainers [16], TiO_2_ nanocontainers [17,18], etc. are reported to have efficient healing agent/inhibitor storing capability. Mesoporous materials also showed excellent hosting ability [19,20,21].

Microencapsulation is an important step in the development of self-healing materials. The desired properties of microcapsules could be tailored via proper control of the preparation parameters. Figure 1 shows the size variation in alkyd resin (derived from palm oil) encapsulated poly(urea-formaldehyde) (PUF) microcapsules that were prepared at a different agitation rate [22]. Many authors [23,24,25,26] mentioned the ability to tailor the required capsules with different thicknesses, morphologies, and sizes before adding them to the paint or to the polymer matrix. Fayyad et al. [24] encapsulate tung oil in a urea-formaldehyde shell. The group used in-situ polymerization to prepare microcapsules with different sizes through changing the stirring rate. Smaller microcapsules had higher corrosion resistance when compared to larger microcapsules [27].

When the capsules are broken due to the damage in coating, the content of these capsules get out and a reaction is activated for self-healing procedure. For example, Figure 2 depicts the rupture of UF microcapsules capsule and release of the reactive epoxy, which acts as a healing agent [12]. The similar action can be arranged for color change or fluorescence light to be produced after the rupture as a sign for the damage. Different parameters [28] should to be identified before tailoring the self-healing coating such as concentrations of containers, concentration of inhibitors, kinetics of additives, diffusion kinetics and environmental conditions.

The porous inorganic materials were also widely practiced as healing agent containers. The proper functionalization of the orifice of mesoporous silica enabled them to act as pH sensitive self-healing agent containers [29]. The organosilyl-functionalization of mesoporous silica containers was carried out with ethylenediamine (en), en-4-oxobutanoic acid salt (en-COO–), and en-triacetate (en-(COO–)_3_) with different organic fillings. It was found that en-(COO–)_3_-type functionalization with content of 0.23 mmol/g showed better pH stimuli self-healing anticorrosive performance, as the functionalized mesoporous silica have high loading (26 wt %) and release (80%) capacities for the inhibitor and stoppage of any possible leaking.

The amount of encapsulatable healing agent or anticorrosion agent in inorganic and polymeric containers is comparatively low. The low amount of healing agent and high manufacturing cost can limit the commercialization of self-healing technology based on such containers for vehicular applications in automobiles and aircraft. To overcome this limitation, new anticorrosive layers that contain core-shell nano- and micro-fibers as healing agent containers were introduced [30]. Electrospinning can be employed to fabricate networks of core-shell nano- and microfibers that were filled with sufficient amount of the healing agent. In such an attempt, Lee et al. [30] prepared core-shell nanofiber coatings on steel substrate with dimethyl siloxane (DMS) as self-healing agent and dimethyl-methyl hydrogen-siloxane (as curing agent) separately in the cores via the dual emulsion electrospinning method. DMS and dimethyl-methyl hydrogen-siloxane were both encapsulated in polyacrylonitrile (PAN) shells. These dual nanofibers were deposited onto a steel substrate. Finally, the nanofiber mats were intercalated with poly(dimethyl siloxane) (PDMS) matrix. The corrosion experiments that were conducted on the manually damaged nanofiber coating proved the self-healing efficiency and corrosion resistance of these coatings. Apart from emulsion electrospinning method, co-electrospinning was also proved to be an efficient method for fabricating encapsulated nanofiber mats [31]. Figure 3 shows a schematic representation of co-axial electrospinning setup. In co-electrospinning, two syringes were simultaneously used to electrospin the dimethylvinyl-terminated dimethyl siloxane (resin) encapsulated PAN and methylhydrogen dimethyl siloxane (curing agent) encapsulated PAN nanofibers, which were collected together by a drum to get intertwined nanofiber mat. This nanofiber mat embedded transparent PDMS coating over steel surface efficiently acted as a corrosion barrier via the healing process by the release of the resin monomer from the fiber cores to heal the corrosion.

Another extrinsic corrosion protection strategy is the layer-by-layer (LbL) deposition of polyelectrolyte and inhibitor nanolayers over the metals surface. In this process, the release of inhibitor is simulated by the change in pH [32]. Andreeva et al. [33] deposited nanolayers of poly(ethyleneimine) (PEI), poly(styrene sulfonate), and an eco-friendly corrosion inhibitor (8-hydroxyquinoline) over aluminum surface. They proposed a self-repairing mechanism, which involves three major steps: (i) pH neutralization activity of the polyelectolytes, (ii) passivation of the cracked surface of the metal by introducing the inhibitors to settle among the polyelectrolyte layers, and (iii) heal the damaged layer.

### 2.2. Green Concept in Self-Healing Coating

Common organic molecules that have heteroatoms are considered to be good inhibitors. These include sulphur, phosphorus, and nitrogen [34]. However, these materials can have negative effects on the environment. The use of natural materials as healing agents or corrosion inhibitors to fill these capsules is considered to be a good green processing approach. The use of green inhibitors was studied by many authors [35,36,37], where corrosion protection can be achieved by the adsorption of organic compounds on the metal surface through constituent polar function groups. Henna leaves extract (HLE) has good protection ability over corrosion. It was reported that HLE with acrylic coating (0.2 wt/vol%) achieved good protection due to the formation of a barrier, nevertheless the resin coating fails with the increase in temperature due to the formation of ionic conducting paths. These conducting routes are formed due to the thermal expansion [37]. 

Many researchers used different types of oils, such as tung oil [38,39], linseed oils [40,41,42,43,44], neem oil [45], and sunflower oil [46], as greener corrosion inhibitors. These natural oils oxidase with atmospheric oxygen and a film of polymerized dried oils are formed for further protection of the coating surface. Among these natural oils, tung oil and linseed oil are the most practiced self-healing agents due to their better drying performance and the ability of emulsification and cross-linking.

Similarly, green alternatives for nanocontainers have been investigated, such as chitosan [47] and cellulose nanofibers [48,49] and different type of naturally occurring clay minerals, including halloysite nanotubes [50], Attapulgite [51], etc. Zheludkevich et al. [47] successfully developed a pre-layer of cerium-doped chitosan that acts as reservoir for corrosion inhibitor for the aluminum alloy 2024 substrate. A complex is formed between the cerium cations and the chitosan. This complex was reported to be responsible for the immobilization and release of the corrosion inhibitor. The mechanism of immobilization and release of self-healing agent in cellulose nanofiber was slightly different from that of chitosan. Vijayan et al. [49] reported that epoxy monomer and amine curing agent were both immobilized on cellulose nanofibers and incorporated to epoxy coating on carbon steel substrate. While epoxy monomer adhered on the surface of cellulose nanofibers, the amine curing agent chemically bonded with fibers. Up on the surface damage, the nanofibers made contact with the water and deformed to release the epoxy monomer, which subsequently reacted with the active functional group in curing agents to recover the damage. The coating based on cellulose nanofibers are proved to be suitable for sea water applications, as these nanofibers become deformed upon contact with water, thereby releasing the adhered corrosion inhibitors into the damaged area. Dong et al. [52] made use the pH-dependent electrostatic interactions between L-valine and halloysite nanotubes to manufacture smart anti-corrosive coating. The shuttling of charge in L-valine between acidic and basic pH has been utilized for the pH based release of L-valine from HNT lumen.

Figure 4 shows the green components used in developing green coatings.

### 2.3. Graphene as Potential Self-Healing Component

With the advancements in graphene research, scientists developed graphene based coatings with good self-healing and anti-corrosive properties. Graphene based materials are commonly used in waterborne coatings as anti-corrosive fillers, due to their excellent barrier properties. In addition to that, various approaches were reported in utilizing the graphene family materials to enhance the self-healing ability. Graphene oxide microcapsules can act as reinforcing healing capsules. They are utilized because they have the following properties: mechanical stability, thin walls, and high loading of healing materials [53], which are important parameters in defining good microcapsules within the coating. Li et al. [53] reported the production of graphene oxide (GO) microcapsules with thin shells thickness of nanometers. The healing material was linseed oil. There was a good compatibility between GO and polyurethane (PU) matrix due to the presence of functional groups on the shell. The production of these mechanically stable microcapsules is easy through th self-assembly of GO sheets. 10 wt% of this microcapsules in waterborne PU coatings was found to heal a scratch with 20 μm in width on hot-dip galvanized steel surfaces. In another technique, graphene oxide/polysterene (GO/PS) containers were produced with 8-HQ inhibitors [54]. The authors controlled the size of the containers from 700 nm to 35 μm. The produced powders can be used in epoxy coatings over mild steel as low cost self-healing anticorrosion materials. Alternatively, Fan et al. [55] used a multilayer structure of GO with branched poly(ethylene mine) (PEI)/ poly(acrylic acid) (PAA) to protect magnesium alloy. In this case, GO acted as both the barrier layer and the multilayers for self-healing. Very recently, Wang et al. used graphene oxide based hybrid structures as healing agent containers [56]. They fabricated graphene oxide-mesoporous silicon dioxide layer–nanosphere structure full with tannic acid healing agent (Figure 5). Such a coating was proven to be effective protection of metal parts used in submarine vehicles from alternating hydrostatic pressure (AHP). Similarly, another research group used graphene oxide@phosphate intercalated hydrotalcite to impart self-healing ability to waterborne epoxy coating [57]. The self-healing ability of graphene oxide@phosphate intercalated hydrotalcite came from the ion exchange between PO_4_^3−^ and Cl^−^ on metal-coating interface.

### 2.4. Other Latest Concepts

The generation of multi-shelled microcapsules boosted the research in self-healing coating. The double shelled microcapsules showed better resistance to salt water, making them new promising materials to be used with waterborne coatings [58]. The formations of double-layered polyurea microcapsule are shown in Figure 6. The inner layer of polyurea microcapsule was formed via interfacial polymerization, and then coated with an outer layer of PUF shell via in situ polymerization.

Similarly, multicore microcapsules were experimented. These materials provided the dual action of self-healing and anticorrosion [59]. The authors showed that the microencapsulation of both corrosion inhibitors and linseed oil in multicore phenol formaldehyde shells could provide successful self-healing anticorrosive coating over polyurethane paint.

Over the last decade, polydopamine has emerged as a versatile material for functionalizing surfaces [60]. Qian et al. [61] developed a new self-healing coating by using polydopamine (PDA) as a pH-sensitive protector for inhibitor loaded in mesoporous silica nanoparticles (MSNs), which overcame the limitation of spontaneous flow of inhibitors molecules from mesopores. The designed PDA-decorated corrosion inhibitor loaded MSNs nanocontainers was used to prepare self-healing water-borne alkyd coatings on mild steel. Benzotriazole, used as a corrosion inhibitor, was loaded at neutral pH, and it facilitated fast release in acidic pH. PDA functioned to govern the discharge of inhibitors and form protective complexes with corrosion products.

Smart containers tailored to specific actions can be used as self-healing materials. For example, containers that respond to light stimulation function in a more controlled manner when compared with those respond to other stimuli, such as pH or temperature, can be produced. Chen et al. [20] developed a reversible light triggered self-healing water-born alkyd coating over aluminium alloy, by incorporating azobenzene- modified hollow mesoporous silica nanocontainers. Upon ultraviolet (UV) irradiation, azobenzene had reversible trans-cis isomerization associated with the decrease in the size of trans to cis isomers, which resulted in expelling molecules out of the nanocontainers to the corrosion site. Upon exposure to visible light, the trans isomer transformed to cis isomer, which prevented the release of corrosion inhibitor. In this way, it is possible to avoid the extra discharge of the inhibitors after healing the corroded area.

## 3. Techniques to Follow-up the Process of Self-Healing in Protective Coatings 

In this section, we describe the common experimental techniques that are used to confirm the self-healing process in proactive coating for metal structures. The working principle and application in self-healing coatings of electrochemical impedance spectroscopy (EIS), scanning vibrating electrode technique (SVET), and scanning electrochemical microscopy (SECM) are discussed in this section.

### 3.1. Accelerated Salt Immersion Test

Accelerated salt immersion test is a preliminary evaluation procedure for self-healing coatings. This is the simplest procedure to follow-up with the self-healing process in protective coating. The scribed coatings are immersed in 10 wt% salt solution, a concentration that is higher than that of normal sea water concentration. The visible progression of corrosion is estimated on definite time intervals. For instance, Huang et al. [62] used accelerated salt immersion test to evaluate the self-healing ability of epoxy coating that contained 10 wt% of polyurethane microcapsules with hexamethylene diisocyanate (HDI) as core material on steel substrate. It was observed that HDI microcapsule loaded coating could resistant corrosion (Figure 7a), while the control coating was extensively corroded (Figure 7b).

### 3.2. Electrochemical Impedance Spectroscopy (EIS)

Electrochemical impedance spectroscopy (EIS) is a routine method for studying the self-healing properties of protective coating for metal [63]. Traditionally, this technique has been used to evaluate of the physicochemical processes that are associated with corrosion on coated substrates. With the development of self-healing coating as an alternative route for anti-corrosion protection, EIS was extended to follow the healing process. EIS was successfully applied to various self-healing coating systems, such as polymers and their composites coatings, metal oxide coatings, coating based on carbonaceous nanomaterials, such as carbon nanotubes (MWCNTs) and graphene etc.

Scribing on the coating surface manually forms defects. EIS consists of a three-electrode system in which the scribed coating acts as the working electrode and normally the Ag/AgCl electrode acts as a reference electrode. A counter electrode is aligned parallel to the coated material to complete the cell. NaCl aqueous solution of desired concentration is used as the electrolyte. The used concentration of NaCl solution depends on several factors, such as the type of coating, area of application, etc. The use of low concentration NaCl helps to estimate the early stage corrosion process more accurately, as it decreased rate of corrosion. The entire experimental setup is schematically shown in Figure 8.

One of the popular representation methods of EIS result is the Bode plot, where log frequency is plotted against both the absolute values of the impedance (|Z| = Z_0_) and the phase-shift. Bode plot for the coated metal surface with a defect is primarily used to probe the self-healing phenomena. Upon creating an artificial defect on the coating surface, the impedance at low frequencies gets suddenly dropped due to the failure of the oxide film and the start of the corrosion process. Once the healing of the defect starts, the impedance recovers its initial values. For instance, Zheludkevich et al. [64] analyzed the Bode plot to follow the self-healing performance of a corrosion inhibitor doped hybrid so-gel coating on aluminium alloy (Figure 9). 8-hydroxyquinoline is used as a corrosion inhibitor in this coating. The authors found that the initially dropped impedance values retrieved the initial value after a short period of 20 min. (Figure 9a). However, continuously decreased low frequency impedance values of undoped single-layer sol–gel coating indicated coating failure without recovery (Figure 9b).

The time that is required to initialize the healing process is an important parameter in judging a self-healing coating. EIS gives evidence of this self-recovery time. The EIS investigation on epoxy coating with epoxy microcapsules applied on carbon steel revealed that the coating was able to heal within 4 h (Figure 10) [14]. This confirmed the microcapsules that were immediately released the healing agent as the cracks propagated through the coating.

An alternative method for quantifying the self-healing process is to fit the EIS data with an equivalent circuit while using data analysis. This method provides some more knowledge regarding the self-healing performance of the coated layers by assigning the impedance spectra to the components of the circuit [62]. The change of components of the equivalent circuits, such as healing resistance (*R_healing_*) and healing capacitance (*C_healing_*) with immersion time, gives direct information about the self-healing process. Hexamethylene diisocyanate (HDI) filled polyurethane microcapsules containing epoxy coating on steel substrate was evaluated for its self-healing performance by fitting the EIS plots with equivalent circuit [62] (Figure 11a). The equivalent circuit of the scratched HDI filled microcapsule containing self-healing coating in EIS measurement is shown in Figure 11b. The value of *R_healing_* increases with the immersion time (Figure 11c), which reflects the self-repairing of coating surface.

### 3.3. Scanning Vibrating Electrode Technique (SVET)

Corrosion and self-healing process in coated metal substrates are initiated over the micro area. This process is usually associated with changes in the electrochemical potential and current density. SVET can probe corrosion initiation by measuring the local corrosion current densities in micro-confined defects. SVET maps the electric field produced above the electrochemically active surface level, and graphically show it as a contour map. This technique provides real time mapping and quantifying of local electrochemical and corrosion events. The use of vibrating reference electrode to measure the electric field improves the resolution and lower minimum measurable signal when compared with a static reference electrode.

The electrochemical cell setup of SVET involves a microelectrode, which is the main part of the SVET. It is immersed in an electrolyte, in which the coated material is fixed with a holder. Figure 12 shows electrochemical cell setup for SVET. A computer program mechanically managed the scanning probe of the microelectrode. The observer can vertically oscillate the probe over the coated material. The results can be recorded as a current density map.

SVET was used to follow the self-repairing in epoxy coating containing corrosion inhibitor doped microgel on aluminum substrate by Latnikova et al. [66]. A maximum increase in current density recorded after 12 h of immersion in SVET current density map confirmed an intense corrosion on control epoxy coating, as shown in Figure 13a–c. At the same time, the epoxy coating containing 5 wt% of corrosion inhibitor doped microgel particles showed no signs of corrosion, indicating the continuous release of 2-methylbenzothiazole (MeBT) from microgel in to the damaged area (Figure 13d–f).

SVET has been successfully used to follow the self-healing process in polyelectrolyte multilayer coating containing an inhibitor on aluminum plates [33]. The current density maps of this polyelectrolyte based coating were compared with that of the SiO_x_/ZrO_x_ sol-gel coating. The current density map of Al surface coated with sol-gel film showed increasing anodic activity with the immersion time ultimately ending up in total failure of metal surface. However, an extremely different behavior was shown by inhibitor loaded polyelectrolyte coated Al alloy surface as neither anodic activity nor corrosion product was observed until 16 h of immersion.

### 3.4. Scanning Electrochemical Microscopy (SECM)

Similar to SVET, scanning electrochemical microscope (SECM) is also an important local electrochemical method for following self-healing with high spatial resolution [67]. The microelectrode probe is the main component in SECM. This probe laterally scans across the coating surface and it is used to monitor processes locally in solutions, and map the corresponding surface reactivity [68]. The size of the probe tip is critical, as it controls the spatial resolution of SECM measurements. SECM facilitates the chemical changes by detecting the ionic species involved in the corrosion process, which is unable to be provided with SVET. Latest literatures report the use of both redox competition mode and a combined redox-competition and negative-feedback mode to monitor healing process. González-García et al. [69] monitored the self-healing phenomena in epoxy coating embedded with capsules containing silyl-ester on aluminum alloys with the help of SECM. They opted a combined mode in SECM that was clearly able to distinguish the oxygen behavior as an active component in the cathodic processes and as a mediator. Figure 14 shows SECM profile of the transition area on aluminum alloy surface with control coating and silyl-treated coating upon immersion in salt solution. The average plateau of more negative current that was observed on silylester-area indicates the absence of cathodic activity. The silyl-ester that was released from capsules regenerates the uniform surface layer to protect the metal from corrosion and consequently more oxygen is available in immersion medium.

## 4. Applications of Self-Healing Coatings

When considering the various technologies available, the development of self-healing coatings depends on the end use applications. Most of the above-discussed self-healing coatings are currently in the product development stage. The introduction of such smart coatings in many modern industries would protect the metal parts from the corrosive environment, improve the performance and ensure safety. The applications include automotive, electronics, aerospace, oil and gas, medical materials, and marine industries. The existing self-healing technologies in the market can repair scratches on paints and coatings in automobiles [70]. Coatings with self-healing ability triggered by sunlight on concrete have been reported [71] and a similar coating over metal substrate would have potential use in outdoor applications. Coatings that use corrosion inhibitors as healing agents would produce cost effective anti-corrosion coatings for infrastructure and oil and gas pipelines. Thin self-healing coatings that were obtained with sol-gel technique are potentially useful for aluminum alloys in aerospace applications. Self-healing coatings are feature smart materials, which can impart aesthetic function in addition to protective functions to automobiles and infrastructure.

Advanced research in this area focuses on integrating multi functionalities on self-healing coatings. Most importantly, properties, such as anti-fouling, super hydrophobic, and anti-friction, can be integrated along with self-healing [72,73]. In this process, different functional materials are used, depending on the required applications. The long time release of healing agents is important for different applications, especially the one that is related to aging protection. In this system, the required treatment is released when needed. For example, in [74], two techniques can be used at the same time: the first technique is the fast release of inhibitors by nanocontainers under the required environment conditions, the other technique is the long time, slow release of inhibitors. Ecofriendly vegetable oils and clay minerals are potential candidates as healing agents and nanocontainers, respectively, for scalable industrial application.

Table 1 summarizes different types of self-healing coatings, their key characteristics, and potential applications.

## 5. Conclusions 

New generation of self-healing protective coatings confirmed their candidature as alternative anti-corrosion coatings. The current review gave a concise discussion on the latest self-healing strategies that were adopted in protective coating technology. Greener coatings avoid the use of solvents and volatile organic coating. The emerging green synthetic approach, role of graphene family members and polydopamine in self-healing coating technology were discussed. We explained both the conventional and advanced techniques that were used to follow the self-healing process in coatings. We also discussed the use of specific new techniques, such as SVET and SECM, in detecting and following the self-healing phenomenon in protective coatings need to be explored more in future. As intensive research is progressing in this field, multifunctional self-healing coatings will be the future of coating technology in marine, aeronautical, and oil and gas industries. The new modified surfaces can offer reduced production cost and prolonged service life.

## Figures and Tables

**Figure 1 materials-12-02754-f001:**
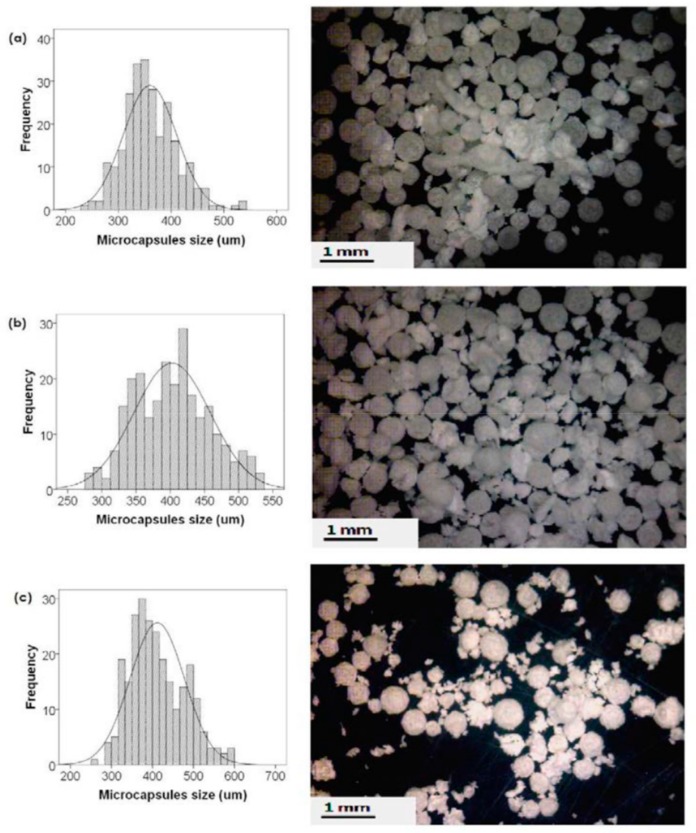
Size distribution (left) and digital microscopic images of alkyd resin encapsulated poly(urea-formaldehyde) (PUF) microcapsules (right), prepared at different agitation rates, (**a**) 500 rpm, (**b**) 400 rpm, and (**c**) 300 rpm. Adapted from [22], with permission from © 2016 Elsevier.

**Figure 2 materials-12-02754-f002:**
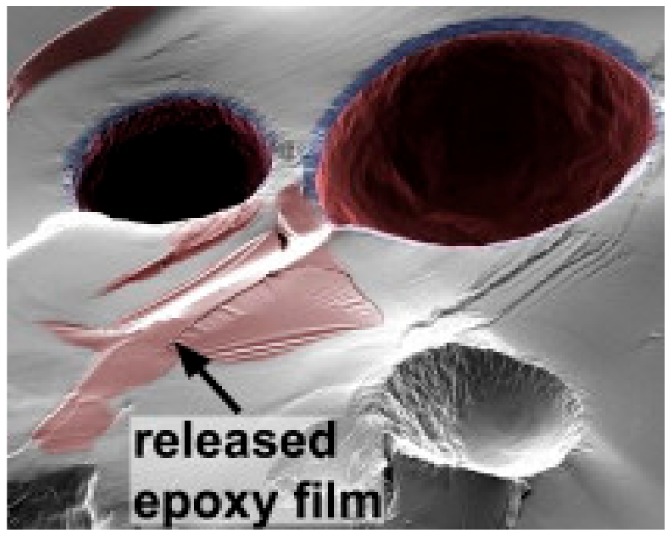
Image of ruptured urea–formaldehyde (UF) microcapsules having both reactive epoxy resin and solvent as the core material. The regions of deposited epoxy film are indicated. Adapted from [12], with permission from © 2009 Elsevier.

**Figure 3 materials-12-02754-f003:**
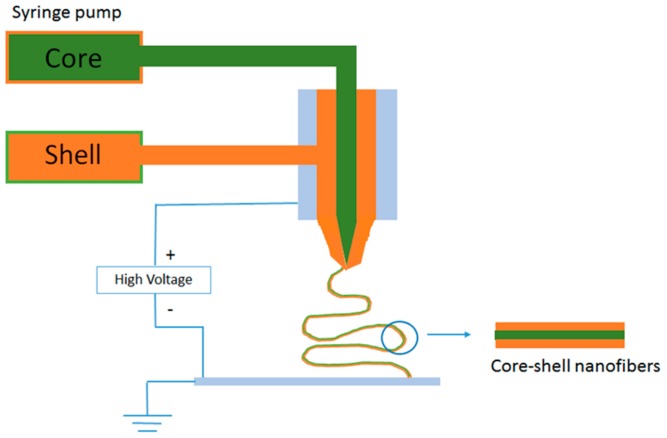
Co-axial electrospinning setup for generating core-shell fibers.

**Figure 4 materials-12-02754-f004:**
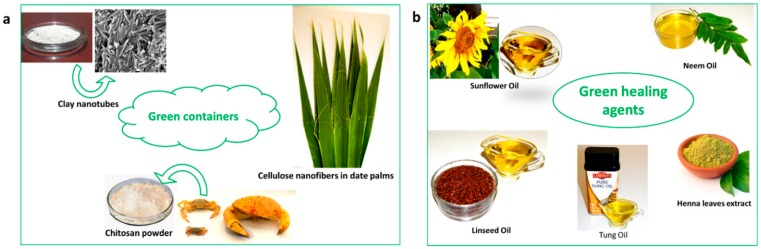
The green components of self-healing anticorrosive coatings.

**Figure 5 materials-12-02754-f005:**
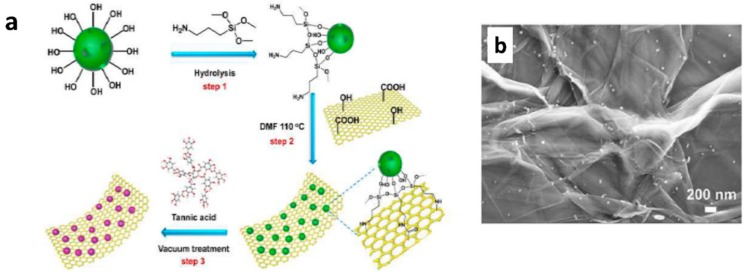
(**a**). Schematic representation of synthesis procedure of graphene oxide-mesoporous silicon dioxide layer-nanosphere; (**b**). Filed emission-scanning electron microscopic (FE-SEM) image of graphene oxide-mesoporous silicon dioxide layer–nanosphere. Adapted from [56], with permission from © 2019 Elsevier.

**Figure 6 materials-12-02754-f006:**
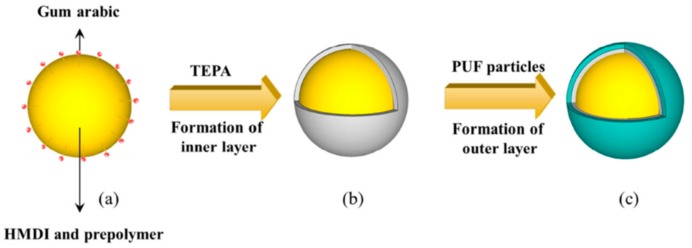
Schematic representation of double-layered polyurea microcapsule formation. Adapted from [58], with permission from © 2016 Elsevier.

**Figure 7 materials-12-02754-f007:**
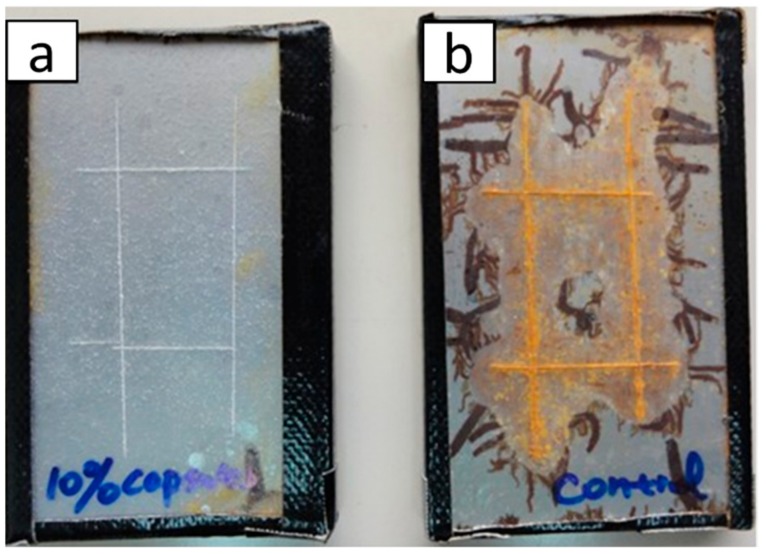
Photographs of steel panel coated with (**a**) 10 wt.% hexamethylene diisocyanate (HDI) microcapsule incorporated coating and (**b**) pure coating after accelerated salt immersion test for two days followed by storage at room temperature for six months. Adapted from [62], with permission from © 2014 Elsevier.

**Figure 8 materials-12-02754-f008:**
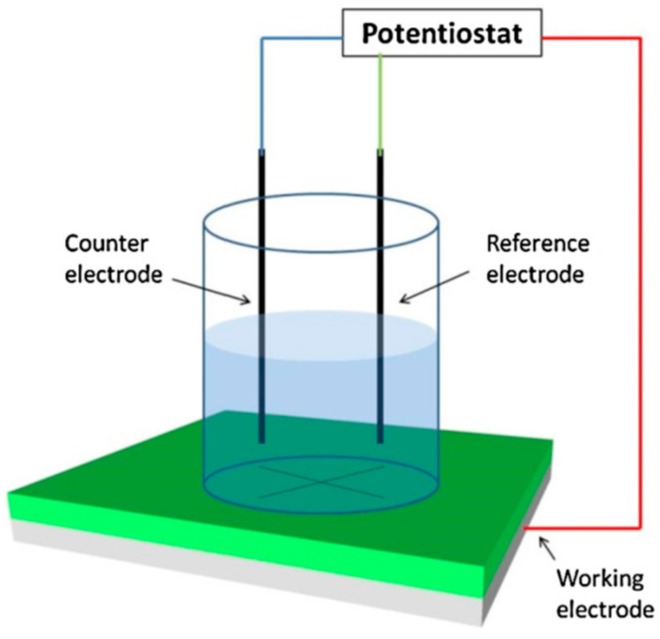
Schematic diagram of the electrochemical impedance spectroscopy (EIS) measurement. Adapted from [62], with permission from © 2014 Elsevier.

**Figure 9 materials-12-02754-f009:**
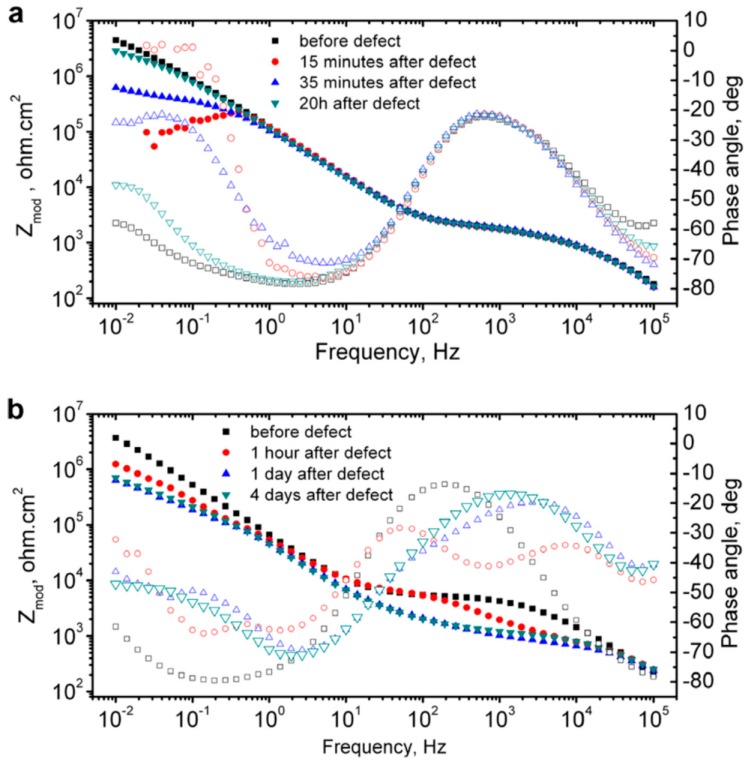
EIS plots of (**a**) inhibitor doped double-layer sol-gel film and (**b**) undoped single-layer sol–gel film during immersion in 0.5 NaCl. Adapted from [64], with permission from © 2007 Elsevier.

**Figure 10 materials-12-02754-f010:**
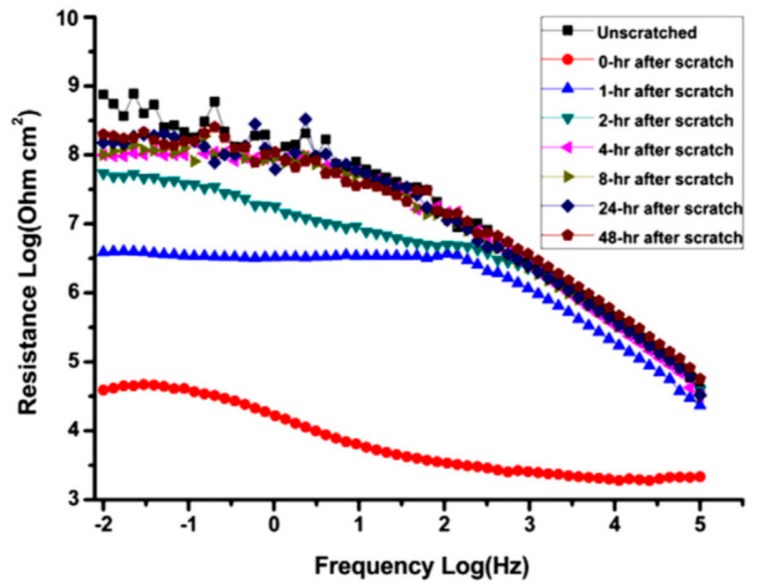
EIS Bode plots of epoxy coatings containing 20% microcapsule during immersion in 12 wt.% NaCl solution at different time intervals. Adapted from [14], with permission from © 2012 Elsevier.

**Figure 11 materials-12-02754-f011:**
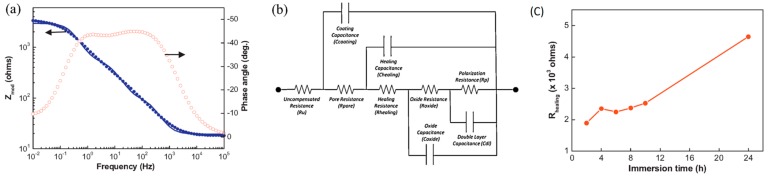
(**a**) Bode plots and fitted curves (solid line) of scratched epoxy coating with HDI filled PU after 8 h of immersion in 1 M NaCl solution, corresponding (**b**) equivalent circuit in EIS measurement and (**c**) healing resistances (*R_healing_*) versus immersion time plot. Adapted from [62], with permission from © 2014 Elsevier.

**Figure 12 materials-12-02754-f012:**
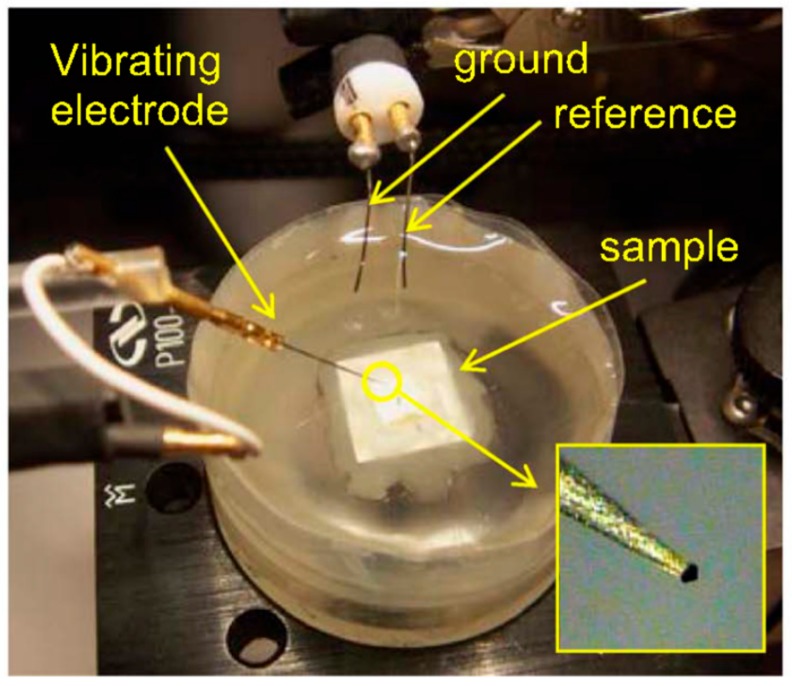
Electrochemical cell setup for scanning vibrating electrode technique (SVET) experiments. Adapted from [65], with permission from © 2017 Electrochemical Society, Inc.

**Figure 13 materials-12-02754-f013:**
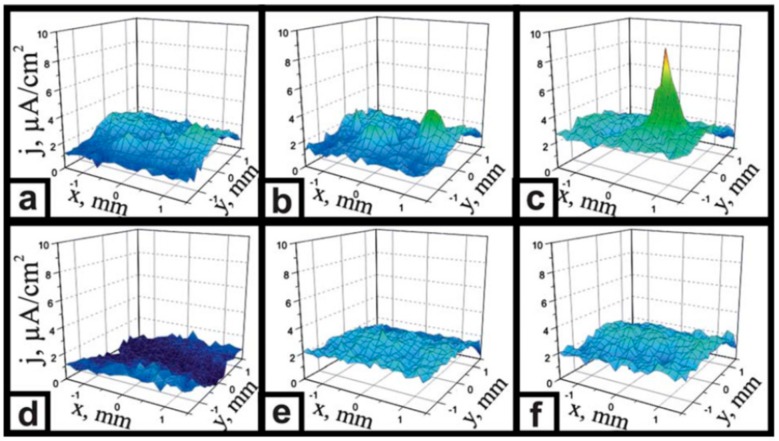
The current density distribution above scratched aluminium substrates covered with (**a**–**c**) control epoxy coating and (**d**–**f**) epoxy coating modified with 5 wt% of MeBT doped microgel particles: **a**,**d**—0 h of immersion, **b**,**e**—6 h of immersion, **c**,**f**—12 h of immersion in 0.1MNaCl solution. Adapted from [66], with permission from © 2012 Royal Society of Chemistry.

**Figure 14 materials-12-02754-f014:**
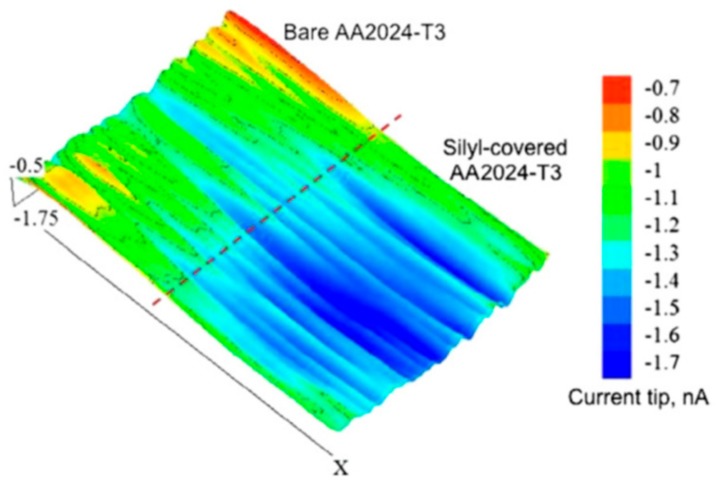
Scanning electrochemical microscopy (SECM) profile of the transition area on aluminium Alloy with control and silyl-treated coatings immersed in 0.05 M NaCl solution for 1 day. Adapted from [69], with permission from © 2011 Elsevier.

**Table 1 materials-12-02754-t001:** Different type of self-healing coating, key characteristics, and potential application.

Sl No.	Type of Self-Healing Coating	Characteristics	Potential Applications
1	Micro/nano polymer capsules to load the healing agent.	Popular self-healing coatings. Preparation of capsules can be tedious. Challenges in stability.	An anticorrosive coating to enhance the durability of metallic structures.
2	Multi-shelled microcapsules to load the healing agent.	Good resistance to salt water.	Waterborne self-healing coatings for automobiles.
3	Porous inorganic materials with functionalized orifices to load the healing agent.	Commercially available porous inorganic materials can be used directly. Controlled release of healing agent.	pH sensitive self-healing coating for metals.
4	Core−shell nano- and micro-fibers as healing agent containers	Sufficiently large amount of healing agent could be loaded in core-shell fibers.	Anticorrosive coating for large scale industrial applications.
5	Layer-by-layer coating to immobilize healing agent/corrosion inhibitor	Thin coating offers long term corrosion protection.	To protect aluminum alloys used for aerospace applications
6	Cellulose nanofibers to immobilize healing agent/corrosion inhibitor	Ecofriendly coating technology.	For submarine applications.
7	Halloysite nanotube as healing agent containers	Economic and green coatings. Halloysite nanotubes act as reinforcing agent for the coating.	Anticorrosive paint for commercial applications.
8	Natural oils as healing agents	Green and economic.	Anti-corrosive metal coatings for scalable industrial applications.
9	Henna leaves extract as corrosion inhibitor	Eco-friendly corrosion inhibitor.	Suitable to protect variety of metals exposed to a wide range of electrolytes.
10	Graphene oxide (GO) based microcapsules as healing agent container	Mechanical stability and high loading capacity.	Protect metal parts used in submarine vehicles from alternating hydrostatic pressure (AHP).

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
