# Peer review of "Self-Repairing Composites for Corrosion Protection: A Review on Recent Strategies and Evaluation Methods"

_materials, 2019, doi:10.3390/ma12172754_

Round 1
Reviewer 1 Report
The manuscript is suitable for publishing after minor revision according to the comments presented below
1. Page 1, Line 40 - The authors should give a proof that the organic coatings are the oldest protective coatings..
2. Figures 1 and 2 are missing.
3. The authors should givethe full names of PUF, TEPA etc. as they give always the full names of the other compounds.
4. Figures 5 and 6 - It is not clear if the figures are from from other article or belong to the authors.
5. Figures 11 - 13 are very unclear and not well visible.
Author Response
1. Page 1, Line 40 - The authors should give a proof that the organic coatings are the oldest protective coatings.
Answer: The statement has been cited from the following chapter; “Taylor, S.R. 22 - The Role of Intrinsic Defects in the Protective Behavior of Organic Coatings. In Handbook of Environmental Degradation of Materials (Second Edition); Kutz, M., Ed.; William Andrew Publishing: Oxford, 2012; pp. 655–672 ISBN 978-1-4377-3455-3.”
This has been cited in the text as reference number: 3. The change has been highlighted in the manuscript with pink colour.
2. Figures 1 and 2 are missing.
Answer: Mistakenly, the figures were wrongly numbered in the previous version of the manuscript. The figure numbers are corrected according to reviewer’s comments.
3. The authors should give the full names of PUF, TEPA etc. as they give always the full names of the other compounds.
Answer: The expansion of the following compounds are provided and the corrections are highlighted in green color.
4. Figures 5 and 6 - It is not clear if the figures are from from other article or belong to the authors.
Answer: Figure 5 and figure 6 (new figure numbers 3 and 4) are not taken from published articles. It has been created by the authors.
5. Figures 11 - 13 are very unclear and not well visible.
Answer: Accordingly, clear images are provided.

Reviewer 2 Report
Materials
Manuscript ID-555901
Title: Self-repairing composites for corrosion protection: A review on recent strategies and evaluation methods.
Reviewer comments:
The authors have produced an interesting review concerning the preparation of organic and hybrid coatings for the corrosion protection of different metals. Some points should be modified. I would suggest publishing it on Materials only after major revisions.
(1) A revision of English language and style is requested: for example line 25 “the metal” should be “metals”; lines 50-51: “With the advancement in coating research, smart coatings which shows stimuli responsive action has been reported by many researchers” should be “With the advancement in coatings research, smart coatings which show stimuli responsive action have been reported by many researchers”, and so on.
(2) Paragraph “2.2 Preparation of BC-PPy, BC-ZnO and BC-PPy-ZnO films”: The authors should re-phrase the paragraph from “In order to study the effect……” to “to the experimental design.” because it is not clear and grammatically not correct. Furthermore, the authors should describe in a clear way the drying conditions adopted (pressure, temperature, time).
(3) lines 43-44, the authors said: “A variant of organic coating, organic-inorganic coatings consist of both the organic component and inorganic fillers (mostly silica) bonded together by the covalent bonds”; the authors should broaden the discussion to other common inorganic fillers used, for example see : Poznyak, S. K., et al. "Preparation and corrosion protective properties of nanostructured titania-containing hybrid sol–gel coatings on AA2024." Progress in Organic Coatings 62.2 (2008): 226-235.; Cui, Xiaokun, et al. "Polydimethylsiloxane-titania nanocomposite coating: fabrication and corrosion resistance." Polymer 138 (2018): 203-210.; Lamaka, Sviatlana V., et al. "Nanoporous titania interlayer as reservoir of corrosion inhibitors for coatings with self-healing ability." Progress in organic coatings 58.2-3 (2007): 127-135.
(4) section 4: Applications of self-healing coatings: The authors should investigate further the different application techniques available, adding relevant references.
(5) In the manuscript, the authors should specify the type of metals that can be used for each example reported.
Author Response
(1) A revision of English language and style is requested: for example line 25 “the metal” should be “metals”; lines 50-51: “With the advancement in coating research, smart coatings which shows stimuli responsive action has been reported by many researchers” should be “With the advancement in coatings research, smart coatings which show stimuli responsive action have been reported by many researchers”, and so on.
Answer: Accordingly, the whole manuscript has been edited for English language.
(2) Paragraph “2.2 Preparation of BC-PPy, BC-ZnO and BC-PPy-ZnO films”: The authors should re-phrase the paragraph from “In order to study the effect……” to “to the experimental design.” because it is not clear and grammatically not correct. Furthermore, the authors should describe in a clear way the drying conditions adopted (pressure, temperature, time).
Answer: It seems the mentioned sentences are not in the current manuscript.
(3) lines 43-44, the authors said: “A variant of organic coating, organic-inorganic coatings consist of both the organic component and inorganic fillers (mostly silica) bonded together by the covalent bonds”; the authors should broaden the discussion to other common inorganic fillers used, for example see : Poznyak, S. K., et al. "Preparation and corrosion protective properties of nanostructured titania-containing hybrid sol–gel coatings on AA2024." Progress in Organic Coatings 62.2 (2008): 226-235.; Cui, Xiaokun, et al. "Polydimethylsiloxane-titania nanocomposite coating: fabrication and corrosion resistance." Polymer 138 (2018): 203-210.; Lamaka, Sviatlana V., et al. "Nanoporous titania interlayer as reservoir of corrosion inhibitors for coatings with self-healing ability." Progress in organic coatings 58.2-3 (2007): 127-135.
Answer: Accordingly, the following text has been incorporated
“Hybrid coatings with titania as the inorganic component has also been reported in sol-gel coatign films and nanocomposites coatings[6,7].” in page 2, line 44, 45 and 46
The reference “Nanoporous titania interlayer as reservoir of corrosion inhibitors for coatings with self-healing ability." Progress in organic coatings 58.2-3 (2007): 127-135” has been cited in page 2, line 79
The changes has been highlighted with red color.
(4) section 4: Applications of self-healing coatings: The authors should investigate further the different application techniques available, adding relevant references.
Answer: Accordingly, the following texts have been added in the manuscript- “Existing self-healing technologies provides the repairing of scratches on paints and coatings in automobiles [71].”
“In authors’ view point, ecofriendly vegetable oils and clay minerals are potential candidates as healing agent and nanocontainers respectively for scalable industrial application.”
The changes has been highlighted with blue color.
(5) In the manuscript, the authors should specify the type of metals that can be used for each example reported.
Answer: As suggested by the reviewer, the type of metals used in different studies were specified and the changes are highlighted with yellow color.

Reviewer 3 Report
This is an interesting and well written review article. Some comments/suggestions that to my point of view can further improve the quality of this manuscript, are given hereafter:
1. The paragraph on ‘applications on self-healing’ coatings feels rather brief in comparison to the other parts of this work, whereas the applicability of these materials is actually one of the most important ones. In addition, can you be more specific on the applications (e.g. which components in the automotive, electronics, aerospace etc. use these coatings?).
2. I would suggest that you add a table to summarize all the coating types presented, their key characteristics and fields of application. There are a lot of information presented in this work and I feel that a summary table will help group all this know-how together, helping the reader to get a better understanding.
Author Response
The authors are grateful to the reviewer for suggestions and comments to improve the quality of the review manuscript.
Q1. The paragraph on ‘applications on self-healing’ coatings feels rather brief in comparison to the other parts of this work, whereas the applicability of these materials is actually one of the most important ones. In addition, can you be more specific on the applications (e.g. which components in the automotive, electronics, aerospace etc. use these coatings?).
A1. ‘Application of self-healing coatings’ has been elaborated accordingly. The newly introduced sentences are highlighted with blue color.
Q2. I would suggest that you add a table to summarize all the coating types presented, their key characteristics and fields of application. There are a lot of information presented in this work and I feel that a summary table will help group all this know-how together, helping the reader to get a better understanding.
A2. We fully agree with the reviewers comment and the following table has been newly introduced in the review manuscript.
Attached is the final version of the paper.

Round 2
Reviewer 2 Report
All corrections have been made